# AraLive: Automatic Reward Adaption for Learning-based Live Video Streaming

## ABSTRACT

Optimizing user Quality of Experience (QoE) for live video streaming remains a long-standing challenge. The Bitrate Control Algorithm (BCA) plays a crucial role in shaping user QoE. Recent advancements have seen RL-based algorithms overtake traditional rule-based methods, promising enhanced QoE optimization. Nevertheless, our comprehensive study reveals a pressing issue: current RL-based BCAs are limited to the fixed and formulaic reward functions, rendering them ill-equipped to adapt to dynamic network environments and varied viewer preferences. In this work, we present AraLive, an automatically adaptive reward learning method designed for seamless integration with any existing learning-based approach in live streaming contexts. To accomplish this goal, we construct a dedicated user QoE assessment dataset for live streaming and customize-design an adversarial model that skillfully aligns human feedback with actual network scenarios. We have deployed AraLive in not only the live streaming but also the classic VoD systems, in comparison to a series of state-of-the-art BCAs. The experimental results demonstrate that AraLive not only elevates overall QoE but also exhibits remarkable adaptability to varied user preferences.

## CCS CONCEPTS

• **Information systems** → **Multimedia streaming**.

## KEYWORDS

Live Video Streaming; Adaptive Reward Learning; Human feedback; QoE Optimization

## 1 INTRODUCTION

The rapid rise in live video streaming is transforming our lifestyle, impacting everything from our social interactions to how we learn and work remotely. This shift is propelled further by technological advancements in areas like volumetric video [24, 25] and AI-generated content (AIGC) [27] streaming, which are significantly enhancing the global uptake of real-time video services. Recent market reports have indicated that by January 2024, video streaming constituted over 72% of all Internet traffic, and this trend is expected to significantly accelerate in further [3].

Video Quality of Experience (QoE) stands as the paramount factor that reflects viewers' perceptual feedback and overall viewing

Permission to make digital or hard copies of all or part of this work for personal or classroom use is granted without fee provided that copies are not made or distributed for profit or commercial advantage and that copies bear this notice and the full citation on the first page. Copyrights for components of this work owned by others than the author(s) must be honored. Abstracting with credit is permitted. To copy otherwise, or republish, to post on servers or to redistribute to lists, requires prior specific permission and/or a fee. Request permissions from permissions@acm.org.

*ACM MM, 2024, Melbourne, Australia*

© 2024 Copyright held by the owner/author(s). Publication rights licensed to ACM.
ACM ISBN 978-x-xxxx-xxxx-x/YY/MM
https://doi.org/10.1145/nnnnnnn.nnnnnnn

satisfaction. Presently, the QoE for real-time video streaming significantly hinges on the effectiveness of bitrate control algorithms (BCAs). These algorithms are designed to dynamically adjust the video bitrate in response to the ever-changing network bandwidth, aiming to maintain an optimal balance between video quality and streaming smoothness. In recent years, there's been a notable shift from traditional rule-based BCAs like GCC and BBR, towards more adaptive, learning-driven approaches, particularly those based on reinforcement learning (RL). Examples include Pensieve [21], Loki [31], R-FEC [18], Jade [14], *etc.* In essence, the central concept behind RL-based BCAs is that the RL agent continuously monitors the network conditions and video QoE, and subsequently determines an optimal bitrate for video streaming, with the objective of maintaining or improving the user QoE. Moreover, this process is significantly influenced by the reward function, which is a critical component of the RL agent.

We have conducted extensive analysis for RL-based BCAs. Our observations reveal that, both in testbed experiments and real-world applications, the reward functions employed by RL-based agents typically follow a formulaic principle. Specifically, they commonly incorporate a variety of video- and network-related metrics, such as video bitrate, frame rate, packet latency, and packet loss rate. Furthermore, these metrics are characterized by diverse hyper-parameters, leading to a predetermined fixed formula. In this work, to demystify the performance of learning-based video streaming, we have performed in-depth measurements (§ 2) over state-of-the-art RL-based video streaming algorithms, such as Loki [31]. Our investigation brings two key findings: *(i)* Formula-based rewards do not consistently align with actual human feedback QoE. Moreover, the probability of mismatches is significantly higher than the anticipation. For instance, a 720P video quality by actual users is likely to be evaluated as either very poor quality (*e.g.*, 270P video) or excellent quality (*e.g.*, 1080P video) by the formula. *(ii)* The rigid formula-based rewards functions fail to accommodate diverse user experiences, which will result in catastrophic QoE.

Inspired by our findings, we are motivated to explore a pivotal question: Can we develop an automatic reward mechanism capable of excelling adaptively across diverse network environments and viewer preferences? To meet this challenge, we design AraLive, an automatically adaptive reward learning mechanism. This solution is designed to be compatible with all existing RL-based approaches used in video streaming scenarios, while also outperforming traditional purely formula-based reward functions.

To meet the above objectives, AraLive introduces two innovative designs. First, recognizing that existing datasets, such as SQoE [9], which are commonly used for VoD streaming training, lack the millisecond-level and fine-grained metrics needed to fully represent the dynamics in live streaming. To bridge this gap, we have constructed a comprehensive user QoE assessment dataset specifically for live-streaming contexts. This dataset includes 4,000 video

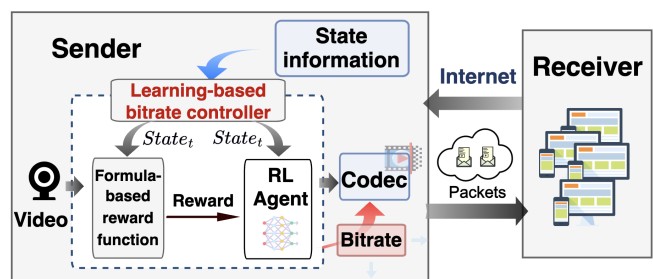

**Figure 1: The workflow of live video streaming based on RL-assisted bitrate control algorithms.**

segments, serving both as a training resource for AraLive and as a valuable asset for the video streaming community. Additionally, we have formulated a dual-weighted reward alignment method to assist AraLive's effective training. Second, we introduce a new adversarial architecture to enable automatic reward adaptation. It features a generator to grasp the implicit relationship between network dynamics and user preferences, subsequently auto-generating the corresponding reward. A discriminator then evaluates these rewards, discerning whether they stem from the generator or human feedback. Through iterative training, AraLive's adversarial model is capable of generating rewards that align with human perception across varying network conditions.

We have validated AraLive's performance in two typical streaming scenarios: live video scenarios and VoD scenarios, benchmarking it against a series of state-of-the-art BCAs. The experimental results show AraLive's ability to maintain consistently high QoE. For instance, AraLive enhances the video bitrate by 4.8% to 9.6%, cuts frame jitter by 11.2% to 11.6%, and reduces stall rates by 3.5% to 50.9%, which shows AraLive's adaptability to different user preferences and network conditions. Surprisingly, AraLive shows even greater gains in the more challenging weak network sessions, which highlights its potential to deploy in practice. Additionally, we present in-depth showcases and perform ablation studies to highlight the advantages of AraLive's model design.

To the best of our knowledge, AraLive is the first work to apply adversarial methods for developing adaptive reward functions aimed at enhancing video QoE for a wide range of individual users. Furthermore, we summarize our key contributions as follows:

- We conduct in-depth analysis and measurements on current learning-based BCAs, identifying their performance is constrained by the reliance on fixed formula-based reward functions (§2).
- We introduce a novel method for aligning actual human feedback QoE with formula-based rewards and present a customized adversarial model, aimed at enhancing AraLive's capability to adapt to actual human QoE (§3).
- We implement AraLive in practical video systems and conduct comparative analyses against a range of baselines. The experimental results demonstrate that AraLive is capable of achieving superior adaptive QoE across various QoE metrics (§4).

## 2 BACKGROUND AND MOTIVATION

### 2.1 Learning-based live video streaming

Recent years, due to the elevated requirements for video quality and the increasingly competitive nature of network bandwidth, ensuring high-quality video QoE has become more complex and demanding. Recent advancements have shown that learning-based BCAs, such as Jade [14], R-FEC [18], Loki [31], OnRL [32], and Pensieve [21], have made significant strides in optimizing QoE. These algorithms utilize ML, particularly RL, to adjust its video bitrate to adapt to dynamic network environments, so as to meet the varying high demands of live video streaming (*e.g.*, maximizing video quality while minimizing the video latency).

In Figure 1, we illustrate the workflow of an end-to-end live video transmission process using the RL-based BCA. In a live video session, the sender's camera will capture video images and encode them into consecutive video frames in real time. These frames are then packetized and transmitted to the receiver over real-world network path. Due to fluctuations of network bandwidth, the sender's BCA will adjust its sending bitrate dynamically to control the amount of data transmitted, thus avoiding network congestion. In particular, the RL-based BCA monitors the instantaneous network states and video QoE, then processes them through a neural network model to determine the next video bitrate. Moreover, it commonly employs a pre-defined reward function to guide the decision-making process of the neural model. As we investigated, the pre-defined reward functions are regulated by a series of hyperparameters. Take the most classic and popular reward function (denoted as $\hat{R}$) in Orca [4, 10] as an example, whose reward is represented as follows,

$$\hat{R} = \left(\frac{\text{throughput} - \zeta \times \text{loss}}{\text{delay}'}\right) \cdot \left(\frac{d_{\min}}{\text{thr}_{\max}}\right),$$
$$\text{delay}' = \begin{cases} d_{\min}, & \text{if } d_{\min} \leq \text{delay} \leq \beta \times d_{\min} \\ \text{delay}, & \text{o.w.} \end{cases}, \quad (1)$$

where we can find that the $\hat{R}$ mirrors the objective of maximizing throughput while minimizing packet delay and loss rate. Specifically, it exhibits a strong correlation with certain hyper-parameters, *e.g.* $\zeta$ and $\beta$, which are coefficients that heavily determine the relative impact of those related metrics.

### 2.2 Inefficiencies of formula-based rewards

To delve deeper into the limitations of formula-based rewards, we aim to leverage actual human feedback QoE as the groundtruth. To this end, we begin by compiling a QoE dataset with human assessment in live video streaming scenarios. In this section, we will first provide a detailed overview of the dataset collection process. We will then present two new insights from comprehensive analysis: *(i)* Formula-based rewards do not consistently align with user actual human feedback; the probability of mismatches is significantly higher than the anticipation. *(ii)* The rigid formula-based rewards functions fail to accommodate diverse user experiences, which results in catastrophic QoE.

**Building a user assessment dataset with 4000 videos.** To accurately capture actual human feedback QoE in live steaming

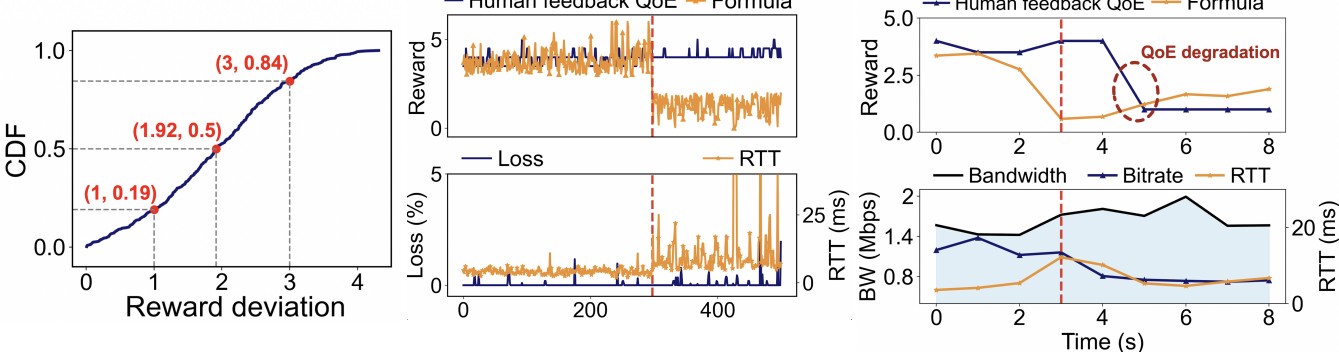

Figure 2: CDF of reward deviation between formula-based reward and human feedback QoE.

Figure 3: Formula-based reward heavily depend on metrics like RTT, loss, which do not align with human feedback QoE.

Figure 4: Showcase of the relationship between rewards and network metrics, and their negative impacts on subsequent QoE.

scenarios, we build a unique user assessment dataset with fine-grained QoE feedback. To achieve this, we first develop a live video system by leveraging the widely used WebRTC framework [1] and also employ a cluster of video segments to serve as the sending video, ensuring comparability across different users and network conditions. Moreover, we use the traffic control (tc) [2] tool to emulate the end-to-end network bandwidth variation, with traces collected from the commercial video system [33]. We also integrate a plugin to record the rewards generated by the formula and add corresponding timestamps.

During the video streaming process, we capture multiple videos transmitted via WebRTC and document the start time of each video session, accurate to the millisecond level. To garner human assessment, we segment the videos and invite tens of volunteers to rate each segment based on the rating ranges of [0, 5]. This procedure yields a comprehensive dataset comprising over 4000 video segments. Each segment is accompanied by ratings from multiple individuals and metrics, such as packet loss, RTT (Round-trip-time), video bitrate, *etc.* Finally, the dataset is composed of a cluster of parameters, including <Video bitrate, RTT, loss, formula-based reward, human feedback QoE>. To resolve the ambiguous boundaries of formula-based rewards, we conduct updates to standardize the range of formula-based rewards, ensuring comparability with human feedback QoE. In particular, as demonstrated in Eq. (2),

$$R_f = \begin{cases} (R_f - \alpha)/(R_{fm} - \alpha) \times R_{hm}, & \text{if } R_f < R_{fm} \\ \frac{R_f - R_{fm}}{\beta - R_{fm}} \times (6 - R_{hm}) + R_{hm}, & \text{else} \end{cases}, \quad (2)$$

where the formula rewards (shorted as $R_f$) are divided into two intervals by using the median value of formula-based rewards (shorted as $R_{fm}$) as the boundary. Then, for each video segment, if its $R_f$ falls within the range of $[0, R_{fm}]$, it is remapped to the new range of $[0, R_{hm}]$, and the remaining values are mapped to $[R_{hm}, 6]$, where $R_{hm}$ represents the median value of human ratings. The parameters $\alpha, \beta$ signify the extreme values of formula-based rewards after outlier removal.

***Insight 1:*** **Existing formula-based reward functions do not align with actual human feedback QoE.** By analyzing the collected QoE assessment dataset, we start by assigning the human feedback QoE for each video segment as $R_h$, and its corresponding

formula-based reward as $R_f$[1]. Next, we calculate the reward deviation $|R_h - R_f|$ between these values. We then depict the CDF in Figure 2, from which we find that, in over 50% of cases, the reward deviation exceeded 1.92 points. This indicates that a video segment rated as 3 (*i.e.*, the average level, which can represent 720P video quality) by users is likely to be evaluated as either very poor quality (e.g., 1, corresponding to 270P video) or excellent quality (*e.g.*, 5, equivalent to 1080P video) by the formula. This significant proportion underlines a substantial misalignment between formula-based rewards and subjective use perceptions. Furthermore, in more than 16% of cases, the difference in rewards exceeds 3. This indicates that for the same video segment, a quality perceived as low by users (*i.e.*, 1 point) may often be deemed high quality (*i.e.*, 4 point) by the formula, and conversely. This further underscores the significant disparity between formulaic results and human feedback QoE.

To investigate the reasons for the deviations between formula-based rewards and human feedback QoE, we further examine key network metrics (*e.g.*, RTT, loss rate) that the reward functions mostly focus on. Specifically, our analysis indicates that the formula-based reward aligns with human feedback QoE when it is low. However, the majority of deviations occur in scenarios where users provide high ratings, but the formula assigns low scores. This pattern suggests that while the formula can accurately reflect user dissatisfaction, it fails to correspondingly recognize higher levels of user satisfaction. Consequently, we illustrate instances where user scores exceed 4 points in Figure 3. By observing the right region of Figure 3, we notice that fluctuations in RTT and loss rate significantly influence the formula-based evaluations. However, minor changes in RTT and loss rate do not impact user QoE as much, which leads to considerable deviations in the assessed rewards.

Furthermore, we provide a showcase (Figure 4) to illustrate the relationships between rewards and network metrics. Specifically, in the first 3 seconds as shown in Figure 4, the formula-based rewards remain lower than the user-perceived QoE due to the increasing trend in network RTT, but the human feedback QoE does not suffer. Subsequently, the RL agent decreases its bitrate in response to the low formula-derived rewards, which results in a downturn in human QoE due to diminished video quality (around 2 seconds

---

[1]The formula-based rewards are obtained from the RL-based BCA—Loki [31].

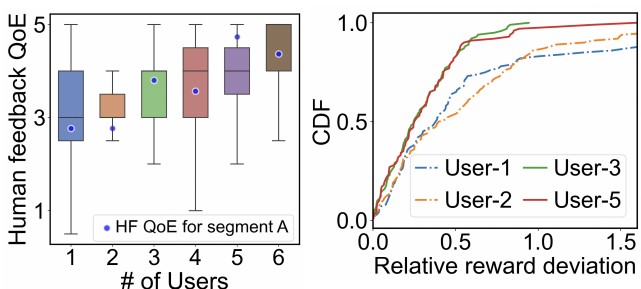

**Figure 5: QoE comparisons over 6 users.**

**Figure 6: CDF comparisons of relative reward deviation.**

later). Although the RTT may gradually improve, the slow recovery causes the bitrate to remain at a low level for about 4 to 6 seconds. As a result, the QoE continues to be lower in subsequent streaming. Hence, we infer that inaccuracies in formula-based rewards can adversely affect the real QoE for a long-term period.

*Insight 2:* **The rigid reward functions fail to accommodate diverse user QoE.** Due to the diverse viewing preferences and requirements among different users, the level of satisfaction with the same video segment can vary significantly. To verify this conjecture, we conduct a study involving a set of approximately 200 video segments, each scored by 6 randomly selected users. The results are visualized through box plots representing the distribution of ratings for each user. As depicted in Figure 5, we find that each user has distinct scoring intervals, confirming the individual variations in satisfaction levels. Specifically, we further examine the rating QoE for a particular video segment (referred to as Segment A). As shown in Figure 5: user-5 and user-3 express high satisfaction, rating Segment A close to or above the 75-th percentile. In contrast, user-2 and user-1 exhibit lower satisfaction, with ratings falling below the 50-th and 25-th percentiles, respectively. *This indicates that the "One Size Fits All" approach of rigid reward functions is impractical to adapt with difference user preferences.*

For the aforementioned 4 users, *i.e.*, user-1, user-2, user-3 and user-5, we plot the CDF of relative reward deviations (shorted as *RRD*), which is calculated as $|R_h - R_f|/R_h$. As depicted in Figure 6, it reveals significant differences among users. Furthermore, a lower RRD suggests that the formula-based rewards more accurately approximate the human feedback QoE. In particular, for the high-satisfaction users, *e.g.*, user-3 and user-5, it exhibited lower mean RRD of 0.293 and 0.315, respectively. Conversely, for the low-satisfaction users, the RRD is 0.665 and 0.595 for user-1 and user-2, respectively. This validates that when the formula-based rewards more closely match the human feedback QoE (*i.e.*, a lower average deviation), the user satisfaction tends to be higher. Conversely, when they diverged, the satisfaction was lower. Hence, the formula-based rewards that aligns closely with user preferences can lead to a better QoE in terms of transmission outcomes. This reinforces the notion that a more effective reward model has a significant impact on enhancing user satisfaction.

## 3 DESIGN

In this section, we first describe the method for aligning actual human feedback QoE with formula-based rewards (§3.1). Subsequently, we detail the adversarial model design of AraLive (§3.2).

### 3.1 Aligning the granularity of actual human feedback QoE and formula-based rewards

The objective of AraLive is to train a model that can autonomously generate adaptive rewards in alignment with actual user QoE. To accomplish this, we plan to leverage the collected dataset (including the formula-based reward and network state information) to train the model. However, we have encountered an issue with the dataset: there is a significant misalignment in the timestamp granularity between human feedback on QoE and the formula-based rewards. Specifically, the former is recorded at a granularity of 2 seconds, while the latter is recorded based on RTCP-derived network state at millisecond-level, typically ranging from 50ms to 200ms.

To solve this issue, the simplest alignment method is averaging all the collected formula-based rewards during the period corresponding to human feedback. However, we find that averaging the formula-based rewards fails to accurately reflect the network state, thereby hindering AraLive's ability to generate adaptive rewards. Let's consider a formula-based reward sequence with timestamps within a 1-second interval: {50ms: 5, 100ms: 5, 150ms: 5, 200ms: 5, 400ms: 25, 700ms: 20, 1000ms: 23}. Then, the average reward is 12.6. However, the average reward of subsequence from 200ms∼100ms is 22.7. The significant discrepancy between the two average values shows that averaging results can dilute the fluctuation of network state, failing to capture the panoramic variability of the network conditions.

In order to provide a more accurate depiction of the network state sequence we apply the duration of each interval as its weight to calculate a weighted mean, and also align it with human feedback QoE in terms of time granularity. The specific method is as follows:

$$R_a = \sum_{i=1}^{n} \frac{t_i - t_{i-1}}{T} \times R_i, \tag{3}$$

where, $R_a$ represents the weighted average of the formula-based rewards, $t_i$ represents the time when the i-th formula-based reward is generated, $T$ stands for the period corresponding to human feedback and $R_i$ is the actual reward value in time $i$.

Furthermore, during the collection of human feedback, we observe a *memory-oriented behavior*. For instance, after viewers have been exposed to low-bitrate videos for some time, they tend to give higher feedback when switching to videos of slightly higher bitrate. Typically, a standard 720P video might receive a feedback score of 3. However, if users switch from a 360P video to a 720P video, the feedback score might increase to 4 or even higher. Conversely, when switching from high-bitrate to low-bitrate videos, viewers often provide lower feedback. These findings indicate that the actual user reward is influenced not only by the instantaneous playback quality but also by the comparative difference to prior video quality.

Inspired by this, our target is to set an indicator to monitor playback quality over time, then define a range interval, with variations beyond this interval signifying substantial changes in video quality. To implement this, we initially use formula-based rewards from the past $n$ periods to approximate the historical quality of playback. Recognizing that the relevance of past data decreases over time, we adjust the weight of these historical rewards to accurately reflect their impact on the present, which is denoted as $\bar{R}_f = \sum_{i=1}^{n} \frac{R_i}{2^{n-i+1}}$.

We then adopt the *standard deviation* $\sigma$ of the formula-based rewards from the past $n$ periods to establish the range interval. Next, we introduce a *Dual-weighted reward function* that determines the need for scaling by comparing the formula-based reward within the current time $R_a$ against $\bar{R}_f$. This function is formulated as follows:

$$R_{dual} = \begin{cases} \alpha \cdot R_a, & R_a > \bar{R}_f + \sigma \\ R_a, & \bar{R}_f - \sigma \leq R_a \leq \bar{R}_f + \sigma \\ \beta \cdot R_a, & R_a < \bar{R}_f - \sigma \end{cases} \quad (4)$$

when $R_a$ exceeds $\bar{R}_f + \sigma$, indicating a notable improvement in playback quality, and users tend to rate higher. In this case, we increase $R_a$ by a factor $\alpha$ (default to 2). Conversely, when $R_a$ falls below $\bar{R}_f - \sigma$, indicating a sudden decline in playback quality. Users tend to indicate lower scores, so we lower $R_a$ by $\beta$ (default as 0.5). For values within $[\bar{R}_f - \sigma, \bar{R}_f + \sigma]$, it suggests the playback quality remains relatively unchanged. Finally, $R_{dual}$ not only aligns with human feedback in time granularity but also meets human habits.

**Heterogeneity processing of different users.** As validated in §2.2, different users have distinct preferences when they rate videos, alongside varying scales of scoring ranges. Therefore, we need to first divide the scoring data of different people into the same range. We posit that the modal value (*i.e.*, *mode*) of ratings provided by a user effectively reflects their overall preferences. Specifically, a rating above the *mode* suggests that a particular aspect of the video surpasses that of typical content. Conversely, a rating below the *mode* suggests a deficiency, such as a lag or a bitrate reduction. Therefore, by using the *mode* of each user's ratings as a benchmark, we standardize the ratings across different individuals to ensure a uniform assessment framework. We initially align the *mode* of different users' ratings and then adjust the remaining ratings according to their deviation from the *mode*, effectively normalizing the scores across different users into a comparable range. It can be formulated as follows,

$$R_h = \begin{cases} \overline{\text{Mode}} + (\text{up}-\overline{\text{Mode}}) \cdot \frac{R_h - \text{Mode}(R_H)}{\text{Max}(R_H) - \text{Mode}(R_H)}, R_h \geqslant \text{Mode}(R_H) \\ \overline{\text{Mode}} - (\overline{\text{Mode}}-\text{low}) \cdot \frac{\text{Mode}(R_H) - R_h}{\text{Mode}(R_H) - \text{Min}(R_H)}, R_h < \text{Mode}(R_H) \end{cases}$$
$$(5)$$

where up and low denote the upper and lower limits of the scoring range. $\overline{\text{Mode}}$ represents the average mode of different users' ratings. $R_h$ denotes a single human feedback, while $R_H$ represents the feedback cluster by a user. $\text{Max}(R_H), \text{Mode}(R_H), \text{Min}(R_H)$ respectively stand for the maximum, mode, and minimum values of a user's ratings.

## 3.2 Adaptive Reward learning design

**Big picture of AraLive model.** In this paper, we aim to devise an automatic reward model that automatically aligns more closely with human feedback QoE by fully leveraging the network state information. The automatic reward model design is illustrated in Figure 8. Specifically, we utilize the network state (*i.e.*, video quality, RTT, and loss rate) and formula-based rewards as inputs, employing actual human feedback as labels for supervised learning. This approach enables the adaptive reward model to learn the implicit relationship between network states and human preferences. Consequently, the model is capable of generating rewards that align with human QoE across varying network conditions.

**Strawman solutions.** Given the dataset's network states are continuous time series, our initial focus is on the RNN model [20].

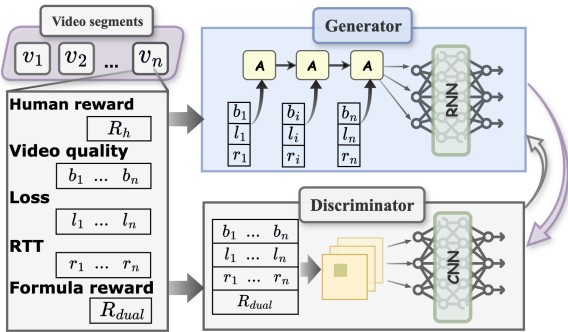

**Figure 7: AraLive's model architecture.**

By feeding sequences of network states into the model and using associated human feedback as labels for supervised learning, we hope to uncover the implicit relationships among input metrics. However, as will be validated in §4.3, we discovered that the RNN model can maintain consistency with the general trends of human feedback. However, it falls short of identifying minor changes in human feedback that result from slight fluctuations in the network state. We also explored utilizing a classic CNN model as the backbone model for AraLive's automatic reward learning. However, as detailed in §4.3, this approach appears prone to overfitting the specifics of network state changes, leading to suboptimal performance.

**AraLive: Achieving automatic reward adaption through adversarial learning.** For AraLive, we hope that it can recognize both human feedback trends and fine-grained network fluctuations. To achieve this goal, we employ the principle of adversarial learning [11] to synergize the capabilities of the two models. Specifically, we designate the RNN model as the generator and the CNN model as the discriminator. During training, the generator aims to mimic human feedback to create adaptive rewards that are indistinguishable from the discriminator; meanwhile, the discriminator evaluates whether the results it receives are actual human feedback or generated by the generator. The RNN model's advantage lies in its ability to rapidly ascertain the approximate scope of human feedback based on network states. Subsequently, it refines the minutiae of human feedback fluctuations through feedback from the discriminator's evaluations. This adversarial process facilitates a dynamic learning environment where both models enhance each other, leading to more accurate emulation of human feedback in AraLive. Through the iterative training process, the generator progressively refines its outputs to closely approximate actual human feedback, thereby minimizing the deviations in its generated results. Concurrently, the discriminator enhances its ability to discern, compelling the generator to learn and adapt to the finer details. Next, we give more details about AraLive's model design (Figure 7).

**AraLive's input and output.** The input of the generator is network state sequence including *video bitrate, loss rate, RTT, formula-based reward*, denoted as $S_t = \{\vec{b_t}, \vec{l_t}, \vec{r_t}, R_{dual}\}$. The model first outputs the probability distribution of rewards and selects the reward with the highest probability as the final result, denoted as $R_g$. The discriminator inputs network state information $S$ and corresponding human feedback or generator's results, the human feedback

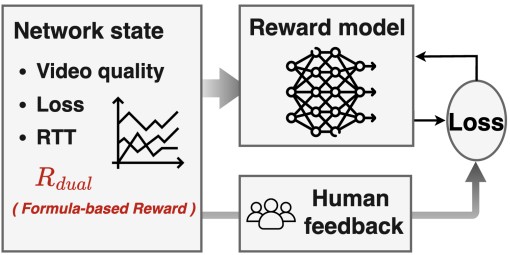

Figure 8: AraLive's automatic reward learning design.

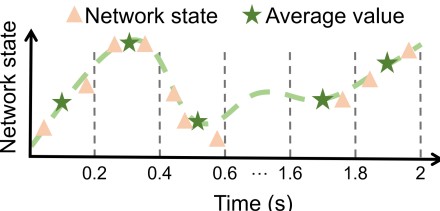

Figure 9: The average values of each 0.2s interval are regarded as the new network state sequences.

corresponds to label 1, and the generator's result corresponds to label 0, outputs the credibility of this corresponding value as human feedback. Moreover, the statistics of the collected dataset show that 60% of the intervals between two adjacent network state sequences are around 200ms, 20% of the intervals are around 100ms, even 5% of them are around 50ms. Uneven intervals may make it more difficult for AraLive to capture the characteristics of network state. Therefore, we want to handle these records, ensuring that network state information is evenly distributed within 2s as well as maintaining detailed fluctuation trends. We find that the minimum interval is 10ms, and the maximum interval is 0.2 s. So, as illustrated in Figure 9, we divide the 2-second period into 10 averaged segments, each with 0.2 s intervals, and the averaged states within each segment are regarded as the new network state.

**AraLive's model architecture.** AraLive is built upon the principles of adversarial learning. In particular, we employ two different model architectures, each tailored to the specific functions *Generator* and *Discriminator*, ensuring that AraLive can effectively learn and adapt to the complexities of video streaming optimization.

*(i) Generator's model architecture.* The input $S_t = \{\vec{b_t}, \vec{l_t}, \vec{r_t}, R_{dual}\}$ is initially transformed into a vector format to facilitate processing by an RNN. Subsequently, this vector is fed into 10 cascaded time steps. The output from each time step, after being activated by the ReLU function, is concatenated and then passed into a 3-layer feed-forward network. This network consists of layers with 256, 128, and 64 neurons, respectively. The ultimate output is a representation of the probability that each score aligns with human feedback.

*(ii) Discriminator's model architecture.* The discriminator first inputs the three metrics, $\vec{b_t}, \vec{l_t}, \vec{r_t}$ to three 1D convolutional layers, separately. Each convolutional layer uses a convolution kernel of 3, with a step size of 1, and an output channel of 3. Subsequently, the outputs of different convolutional layers, $R_{dual}$, and reward value (human feedback or generator's results) are concatenated into a 1D vector, which is then processed by a 3-layer feed-forward network. The output represents the confidence level of the reward value.

**Loss function of AraLive.** To leverage the discriminator's assessments for guiding the generator, we employ the discriminator's evaluation of the generator's outputs as a loss function. More specifically, we employ the discriminator to provide a confidence level for each potential reward value, thereby generating a probability distribution of rewards. Following this, the difference between the probability distributions generated by the generator and the discriminator is calculated, which serves as the loss function for the

generator and is defined as follows:

$$L_G = \sum_x G(S) \cdot \left| \log \frac{\min(G(S), D(x))}{\max(G(S), D(x))} \right|, \quad (6)$$

$G$ represents the generator, $S$ represents the input of the generator, $x$ is a reward value. $D$ represents the discriminator, which outputs the confidence level of $x$. When the probability distribution output by generator is closer to discriminator, the loss value is smaller, which indicates that the results generated by the generator are closer to human feedback.

The input of discriminator can be divided into two parts: actual human feedback and the result of generator. The discriminator should be able to distinguish it. In particular, when the input is human feedback, the output should be as close as possible to 1, and when the input is generator's result, the output should be as close as possible to 0. Therefore, the loss function of the discriminator is designed as follows:

$$L_D = \alpha \cdot (y \cdot \log D(x) + (1-y) \cdot \log(1-D(x))) + \beta \cdot \left| \frac{R_h - x}{1 - D(x)} \right|, \quad (7)$$

where $x$ represents the reward value we need to determine the confidence level, and $y$ represents the label corresponding to $x$. $R_h$ represents human feedback reward. The first item calculates the difference between the output result and the label, and the second item is to make the model output different results for human feedback and the result of the generator, imposing penalties to make the confidence in the result of generator closer to 0. $\alpha$ and $\beta$ represent the ratios of the two items in the overall loss function.

## 4 EVALUATION

In this section, we first detail the experiment methodology, then evaluate AraLive's benefits mainly in the live video streaming scenarios, comparing it against a series of state-of-the-art benchmarks. Finally, we delve deeper into the key modules of AraLive to understand how they automatically generate human-like rewards.

### 4.1 Experiment methodology

**Baselines:** *(i)* Loki [31], an RL-based bitrate adaptation algorithm, innovatively combines learning-based and rule-based BCAs for optimizing live streaming performance. *(ii)* OnRL [32], a PPO-based [22] BCA, operates on the principle of dynamically updating bitrates in an online manner, guided by the instantaneous state of the network. In particular, we have substituted Loki's and OnRL's formula-based reward functions as AraLive, and they are renamed as *Loki*-AraLive and *OnRL*-AraLive. The live streaming system is built upon WebRTC framework [1]. We also validate AraLive

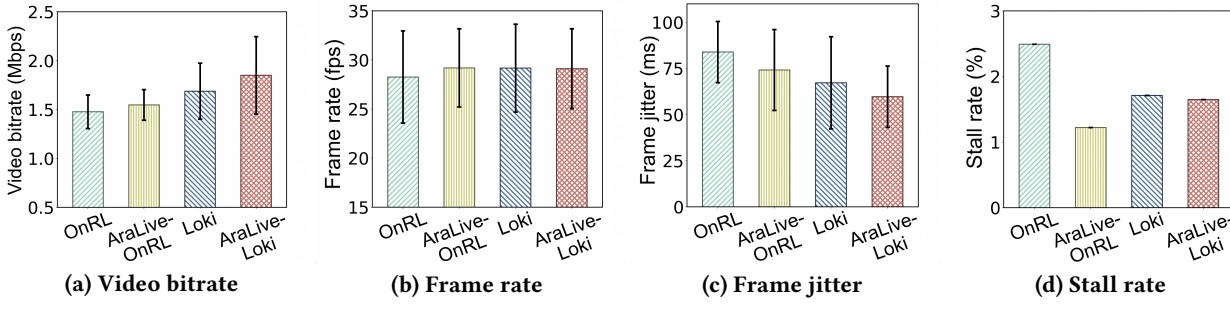

**Figure 10: Overall QoE comparisons in live video streaming scenarios.**

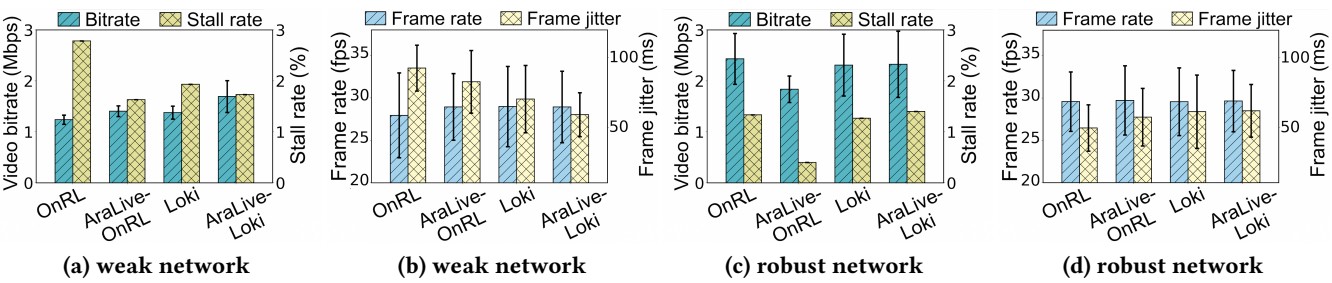

**Figure 11: Network breakdown comparison in live video streaming scenarios.**

in the traditional VoD scenarios, comparing it with Pensieve [21], Fugu [28], and MPC [30]. The detailed comparisons are described in the *Supplementary Materials*.

**Dataset, scales and evaluation metrics:** We utilize the collected user assessment dataset described in §2.2 to evaluate AraLive's performance. Specifically, 80% of the dataset is allocated for training AraLive's model, while the remaining 20% is reserved for testing and validation. For the live video streaming scenarios, we conduct over 100 sessions, with each session lasting over 20 minutes, amounting to a total of over 45 hours of streaming. Moreover, our evaluation primarily focuses on application layer metrics such as *video bitrate*, *frame rate*, *fame jitter*, and *stall rate*.

## 4.2 Overall performance comparison

Figure 10 illustrates the QoE metrics between AraLive and its benchmarks in live streaming scenarios, which reveals notable improvements when AraLive is utilized as the reward function. For example, when compared to *OnRL*, *OnRL*-AraLive achieves a 4.8% increase in bitrate, 3.3% enhancement in frame rate, 11.6% reduction in frame jitter, and 50.9% decrease in stall rate. Similarly, *Loki*-AraLive shows a 9.6% boost in bitrate, 11.2% decrease in frame jitter, and a 3.5% reduction in stall rate while maintaining a comparable frame rate to *Loki*. Among these improvements, *Loki*-AraLive stands out in nearly all metrics than *OnRL*-AraLive, which can be attributed to *OnRL*-AraLive adopting a more conservative approach in its decision-making process, opting for a lower bitrate to minimize stall rate. We've also assessed the overhead implications. Specifically, compared to *OnRL*, *OnRL*-AraLive incurs a 0.4% increase in memory usage. Similarly, compared to *Loki*, *Loki*-AraLive

has a 0.4% increase in memory consumption, which is considered acceptable for typical neural network model deployment.

**Network condition breakdown.** We categorize live streaming sessions into two groups based on their varying network conditions: *robust network* for sessions with consistent, high throughput (taking a fraction of 35%); and *weak network* for those experiencing lower throughput and significant variability (taking a fraction of 65%). The comparisons are shown in Figure 11. Our analysis shows that AraLive significantly bolsters OnRL and Loki's performance, notably in lowering stall rates and frame jitter without negatively impacting frame rate. *This improvement is particularly remarkable in weak network sessions, highlighting* AraLive's *effectiveness in optimizing QoE under challenging conditions.* Specifically, in weak network sessions, when compared to OnRL, OnRL-AraLive achieves a 13.3% increase in frame rate, a 10.6% reduction in frame jitter, a significant 41.3% decrease in stall rate, and a modest 3.6% improvement in bitrate. Similarly, Loki-AraLive outperforms Loki by increasing the bitrate by 22.9%, and reducing the stall rate and frame jitter by 10.3% and 16.0%, respectively. This performance enhancement is attributed to AraLive's ability to more accurately gauge the appropriateness of current bitrate decisions for the network's dynamic fluctuations, enabling OnRL and Loki to make more informed and effective adjustments. In robust network sessions, AraLive also helps OnRL and Loki achieve a certain degree of QoE improvements.

## 4.3 In-depth understanding AraLive

**Performance of AraLive's adaptive reward.** We further evaluate AraLive's abilities to align with actual human feedback QoE. Initially, we calculate the deviations of AraLive's output $R_g$ and the

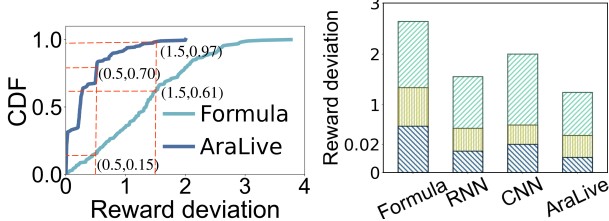

**Figure 12: The reward deviation generated by `AraLive` or formula-based function.**

**Figure 13: The reward deviation comparisons of different models.**

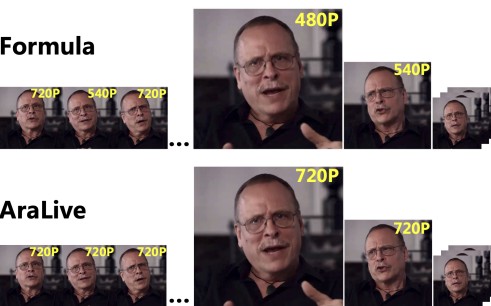

**Figure 14: `AraLive` consistently delivers high video quality than the formula-based rewards.**

formula-based reward $R_f$ as mentioned in § 2.1, in comparison to actual user feedback QoE $R_a$. As depicted in Figure 12, the ratings $R_g$ generated by `AraLive` closely mirror user feedback, whereas the formula-derived $R_f$ exhibit a significant deviation. Specifically, 70.1% of the deviations from `AraLive` are within 0.5, and 97.3% are within 1.5. In contrast, only 15.3% of formula-based reward errors are within 0.5, and 61.2% fall within 1.5. Figure 14 showcases the visualization of `AraLive` compared to formula-based rewards, which illustrates that `AraLive` consistently delivers high video quality.

**Comparisons of different NN structures.** Figure 13 shows the performance of using RNN and CNN for human feedback adaption. We can observe that both RNN and CNN models can better fit the human feedback than the purely formula-based reward function but still underperform than `AraLive`. For instance, 50% of the RNN model's results have an error exceeding 0.514, and for the CNN model, 50% of results have an error greater than 0.532. In comparison, `AraLive` demonstrates a more refined accuracy, with 50% of its results having an error of just 0.239. Moreover, at a 95% confidence level, the error in `AraLive`'s results is reduced by 19.8% and 37.7% compared to the RNN and CNN models, respectively.

## 5 RELATED WORK

**Video streaming transmission.** Driven by the rising demand for instant content delivery, video streaming has evolved from traditional on-demand (VoD) to encompass real-time and live video streaming. Unlike VoD, live video streaming requires instant generation and transmission, necessitating agile processing to handle millisecond–level lower latency [19, 34]. Traditionally, VoD services commonly adopt the Adaptive Bitrate (ABR) algorithms to manage the video bitrate [15]. Notable examples include Cubic [12], BBR [5], MPC [30], Remy [26], Pensieve [21] and TCP-RL [17]. In recent years, with the advancement of Internet instruction and stringent low-latency streaming requirements, many studies have shifted to the Real-Time Communication (RTC) protocol [23], employing the widely adopted WebRTC framework [1] for streaming. The state-of-the-art algorithms include GCC[6], Orca[4], OnRL [32], Loki[31], R-FEC[18] *etc*. In this paper, we have validated `AraLive`'s performance not only in classic VoD services but also in the latest live streaming scenarios.

**Learning-based bitrate control algorithms.** The rapid advancement of machine learning (ML) [13, 16] has led to the dominance of learning-based bitrate control algorithms in managing and controlling video bitrates. Remy [26] was pioneering in integrating ML, specifically a table-based Markov model, into ABR algorithms. This initiated a wave of learning-based solutions for bitrate control in both VoD and live streaming contexts, including PCC [7], PCC-Vivace [8], TCP-RL [17], Pensieve [21], Indigo [29], Fugu [28], Orca [4], Onrl[32], Loki [31], *etc*. The core components of these algorithms can be categorized into two main parts. The first is to customize-design a specific neural network model, which utilizes the history network state as its input to make bitrate decisions. The second involves the configuration of the reward functions, which directs the decision-making process of the neural network. Typically, this function is a linear transformation of crucial network or QoE metrics. For example, the objective of Orca's reward [4, 10] is designed to minimize loss and latency while maximizing throughput based on a linear function. However, this approach cannot adequately account for the nuanced aspects of transmission QoE and also cannot adapt well to user-specific preferences (as validated in §2). The most latest work, Jade [14], is the first attempt to use human QoE as the reward function in RL models. Yet, `AraLive` distinguishes itself from Jade in two key aspects: *(i)* Jade is specifically tailored for VoD scenarios using ABR algorithms, making adjustments at the chunk level with second-level granularity. In contrast, the adjustments in live streaming of `AraLive` need millisecond-level. *(ii)* Jade primarily addresses QoE considerations at the application layer, whereas `AraLive` extends its focus to encompass both the application and transport layers, necessitating a unique model design for `AraLive`. More detailed comparisons of `AraLive` with representative ABR algorithms are shown in the *Appendix materials*.

## 6 CONCLUSION

In this paper, we have designed `AraLive`, an automatic reward learning solution that can adapt with diverse network environments and meet diverse viewer preferences. Practical deployment and massive evaluation on real video system demonstrate that `AraLive` can significantly enhance the QoE in comparison with state-of-the-art bitrate control algorithms. This advancement effectively addresses common concerns regarding the reliability of learning-based solutions, thus paving the way for their broader integration into commercial real-time video streaming systems. We think that the idea of `AraLive` signifies a significant step toward the widespread adoption of learning-based algorithms in the multimedia and networking community.

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
