# OpenReview forum: "AraLive: Automatic Reward Adaption for Learning-based Live Video Streaming"
_acmmm.org/ACMMM/2024/Conference — MM2024 Poster_

### Official Review · Reviewer_drXG · 2024-05-11

**Rating:** 4
**Confidence:** 4

**Summary:**

This paper primarily investigates the inconsistency between the reward function and QoE in reinforcement learning-based live streaming bitrate control algorithms. By establishing a QoE assessment dataset and employing adversarial learning, this paper develops Aralive, an adaptive reward learning method, and combines Aralive with various bitrate control algorithms.

**Strengths:**

1. Addressing the inconsistency between the reward function and user experience in reinforcement learning-based live streaming bitrate control algorithms is crucial for constructing better bitrate control algorithms and improving user experience.
2. The paper is well-structured, and the explanations of the motivation and methods are very clear.
3. Aralive is tested not only in live streaming scenarios but also in video-on-demand scenarios.

**Limitations:**

1. The paper uses the reward from Loki, but there are many other types of rewards available. There are also numerous works studying MoS modeling, but the paper lacks discussion and comparison with these related studies.
2. Regarding the novelty of this paper, it mentions that the difference between Aralive and Jade is that Aralive is used for live streaming. However, Aralive does not seem to have any specific design tailored for live streaming scenarios. Moreover, Jade can also be applied to live streaming by including loss rate in the input, just as Aralive can be used for video-on-demand. Therefore, the comparison with Jade should be present in the experimental section, at least for the VoD part.
3. In Section 2.2, it is unclear why $R_f$ is mapped to [0,6] while the MoS range is [0,5]. Furthermore, this section mentions "The rigid reward functions fail to accommodate diverse user QoE," but there is no method described in the subsequent text to address this issue. Equation (5) linearly maps MoS to [low,up] by comparing with the mode, but this actually mixes the requirements of different types of users instead of providing personalized treatment.
4. $R_{hm}$ in Section 2.2 is not explained, $R_f$ represents two different meaning, and the same applies to $R_h$, $\alpha$ and $\beta$. Moreover, there is no discussion on how to set $\alpha$ and $\beta$ in the paper.

**Suitability:**

3

---

### Official Review · Reviewer_QwaN · 2024-05-18

**Rating:** 6
**Confidence:** 3

**Summary:**

This paper proposes AraLive, an automatic reward adaptation method for training learning-based bitrate algorithm.

**Strengths:**

1. An important topic. Video streaming now is the dominant content type in networking traffics. Understanding how users consume and perspective videos is critical both in academia and industry.
2. Comhensive motivation and measurement. The main idea of this paper is the disalignment between formularized reward and human-based reward (or QoE). The paper provides good insights by analyzing real user feedback.
3. Good evaluation. This paper compares with multiple and strong baselines.
5. Good organization. This paper writes well and easy to follow.

**Limitations:**

I only have several small questions.
1. What does the video segment mean? Does it mean chunk?
2. Why it is 6 in Eq.(2)? How to choose it?
3. Is there any underlying reason that can express the insensitivity of human-based QoE for different QoS metrics?
4. The motivation shows that QoE may not align with QoS, but the the final result shows that AraLive achieves better QoS. How to explain it?

**Suitability:**

3

---

### Official Review · Reviewer_cMRU · 2024-05-25

**Rating:** 3
**Confidence:** 3

**Summary:**

This paper propose a learning-based reward design method for the training of DRL agent in WebRTC bitrate control scenarios. Through in-depth analysis and measurements on existing bitrate control algorithms, the authors argue that they have identified two weaknesses of the current formula-based reward function. A customized GAN-based reward model is then proposed to enhancing the QoE performances for the existing bitrate control algorithms.

**Strengths:**

1. The in-depth analysis and measurements on existing bitrate control algorithms w.r.t the reward function are motivated.
2. It is a reasonable method to design a reward or QoE metric for Web-RTC scenarios.

**Limitations:**

1. The fairness issues in the experiments should be clearly explained and resolved. What is the reward formula used for training Loki and OnRL? Is it the same as the eq.~(1)? The performance comparison in terms of the total QoE should be provided. This is because the policy trained with RL is optimized to maximize the return, which refers to the discounted long-term rewards.
2. Furthermore, the experiments w.r.t VoD scenarios are not persuasive. The reward designs for VoD and RTC are totally different. Why the Pensieve’s policy that is trained for optimizing a trade-off between maximizing the bitrate and minimizing the rebuffering and smoothness penalties will receive significantly performance declines both in these three components, compared to Pensieve-Aralive?
3. The abstract should be rephrased, the key contributions of this paper should be elaborated on the abstract. What is the difference between the perceptual QoE assessment method and the work of this paper?
4. The effectiveness of the collected human assessment dataset needs more discussion. Has it been open-sourced?

**Suitability:**

3

---

### Official Review · Reviewer_tS7m · 2024-05-28

**Rating:** 4
**Confidence:** 3

**Summary:**

The paper introduces AraLive, a novel approach to optimizing user Quality of Experience (QoE) in live video streaming by addressing the limitations of traditional formula-based QoE functions. The authors acknowledge that existing methods fail to accurately predict user experience, leading to the creation of a new dataset and the training of a neural network-based QoE assessment function. This function is designed to enhance the performance of video transmission algorithms by dynamically adjusting to user feedback and network conditions.

**Strengths:**

The authors have made a significant contribution by collecting a new dataset that captures fine-grained user feedback and network conditions. This dataset serves as a valuable resource for training a neural network model that can more precisely assess QoE.

**Limitations:**

1. While the motivation behind this work is clear, the paper's main contribution lies in the creation of a new dataset and a corresponding deep learning approach for QoE prediction. The novelty of the approach is somewhat diminished by the fact that the core idea of using neural networks to predict QoE is not new. The paper would benefit from a more profound exploration of how AraLive differs from existing solutions and why these differences are advantageous.

2. The paper employs adversarial learning to improve the accuracy of the QoE assessment model. However, it does not conclusively demonstrate the necessity of this approach for enhancing accuracy. The experimental section lacks a clear comparison that shows adversarial learning is superior to other potential methods. For instance, it is not evident whether a CNN or RNN baseline model, if trained with adversarial techniques, would yield similar or better results.

3. The paper proposed an RNN-based QoE prediction network and but doesn't address the potential computational overhead it introduces to the original Bitrate Control Algorithm (BCA). Will the increased complexity of incorporating an RNN-based prediction network affect the real-time decision of the RTC system?

**Suitability:**

3

---

### Meta-Review · Area_Chair_DSTn · 2024-07-02

**Recommendation:** Accept (Poster)
**Confidence:** 4

**Metareview:**

As per the reviewer's feedback:

The paper introduces AraLive, a novel approach to optimizing user Quality of Experience (QoE) in live video streaming by addressing the limitations of traditional formula-based QoE functions. The authors create a new dataset and train a neural network-based QoE assessment function, aiming to enhance video transmission algorithms' performance by dynamically adjusting to user feedback and network conditions.

Pros:
* The authors have collected a new dataset capturing fine-grained user feedback and network conditions. This dataset is a valuable resource for training a neural network model that can more precisely assess QoE.
* Addressing the discrepancy between formula-based reward functions and human-perceived QoE is crucial for improving user experience in video streaming. The paper's motivation and goals are well-aligned with practical needs in multimedia processing.

Cons
* While the creation of the dataset and the application of neural networks for QoE prediction are novel contributions, the core idea of using neural networks for this purpose is not entirely new. The paper would benefit from a more detailed exploration of how AraLive differs from existing solutions and why these differences are advantageous.
* The paper employs adversarial learning to improve the accuracy of the QoE assessment model but does not conclusively demonstrate the necessity of this approach. A clear comparison with other potential methods, such as baseline CNN or RNN models, would strengthen the argument.
* The potential computational overhead introduced by incorporating an RNN-based prediction network into the original Bitrate Control Algorithm (BCA) is not addressed. The impact of this increased complexity on real-time decision-making in the RTC system should be discussed.
* The abstract should be rephrased to elaborate on the key contributions of the paper. The differences between the perceptual QoE assessment method and the work in this paper need clarification.
* The effectiveness of the collected human assessment dataset requires more discussion. It would be beneficial to mention whether the dataset has been open-sourced for the research community.

Final Rating Justification:
The paper addresses an important issue in optimizing QoE for live video streaming and presents a novel dataset and method with a comprehensive evaluation. However, there are several areas where the paper could be strengthened, particularly in demonstrating the more clearer the novelty of the approach and justifying the use of adversarial learning.